# Hereditary Ovarian Cancer: Towards a Cost-Effective Prevention Strategy

**DOI:** 10.3390/ijerph191912057

**Published:** 2022-09-23

**Authors:** Aruni Ghose, Anita Bolina, Ishika Mahajan, Syed Ahmer Raza, Miranda Clarke, Abhinanda Pal, Elisabet Sanchez, Kathrine Sofia Rallis, Stergios Boussios

**Affiliations:** 1Department of Medical Oncology, Barts Cancer Centre, St. Bartholomew’s Hospital, Barts Health NHS Trust, London E1 1BB, UK; 2Department of Medical Oncology, Mount Vernon Cancer Centre, East and North Hertfordshire NHS Trust, London SG1 4AB, UK; 3Department of Medical Oncology, Medway NHS Foundation Trust, Gillingham ME7 5NY, UK; 4Department of Medical Oncology, Clatterbridge Cancer Centre, Clatterbridge Cancer Centre NHS Foundation Trust, Liverpool CH63 4JY, UK; 5Department of Medical Oncology, Apollo Cancer Centre, Chennai 600001, India; 6Department of Internal Medicine, St. Thomas’ Hospital, Guy’s and St. Thomas’ NHS Foundation Trust, London SE1 7EH, UK; 7Department of Internal Medicine, Royal London Hospital, Barts Health NHS Trust, London E1 1BB, UK; 8Department of Internal Medicine, IQ City Medical College and Narayana Hospital, Durgapur 713206, India; 9Cancer Research Institute, Beth Israel Deaconess Medical Center, Harvard Medical School, Boston, MA 02115, USA; 10Centre for Experimental Cancer Medicine, Barts Cancer Institute, Queen Mary University of London, London E1 4NS, UK; 11Faculty of Life Sciences & Medicine, School of Cancer & Pharmaceutical Sciences, King’s College London, London WC2R 2LS, UK; 12AELIA Organization, 9th Km Thessaloniki—Thermi, 57001 Thessaloniki, Greece

**Keywords:** ovarian cancer, genetic testing, BRCA, risk-reducing surgery, guidelines, cost effectiveness

## Abstract

Ovarian cancer (OC) is the most lethal gynaecological malignancy. The search for a widely affordable and accessible screening strategy to reduce mortality from OC is still ongoing. This coupled with the late-stage presentation and poor prognosis harbours significant health-economic implications. OC is also the most heritable of all cancers, with an estimated 25% of cases having a hereditary predisposition. Advancements in technology have detected multiple mutations, with the majority affecting the *BRCA1* and/or *BRCA2* genes. Women with *BRCA* mutations are at a significantly increased lifetime risk of developing OC, often presenting with a high-grade serous pathology, which is associated with higher mortality due to its aggressive characteristic. Therefore, a targeted, cost-effective approach to prevention is paramount to improve clinical outcomes and mortality. Current guidelines offer multiple preventive strategies for individuals with hereditary OC (HOC), including genetic counselling to identify the high-risk women and risk-reducing interventions (RRI), such as surgical management or chemoprophylaxis through contraceptive medications. Evidence for sporadic OC is abundant as compared to the existing dearth in the hereditary subgroup. Hence, our review article narrates an overview of HOC and explores the RRI developed over the years. It attempts to compare the cost effectiveness of these strategies with women of the general population in order to answer the crucial question: what is the most prudent clinically and economically effective strategy for prevention amongst high-risk women?

## 1. Introduction

Globally, ovarian cancer (OC) is the eighth most common cancer among women and the eighteenth most common cancer overall [1]. With around 7500 new cases every year and 5% of cancer-related deaths, OC is the sixth most common cancer as well as sixth most common cancer-related death among women in the United Kingdom (UK) [2]. In the United States of America (USA), it is the fifth most common cancer-related death [3]. OC has a lifetime risk of 1 in 78 and probability of mortality of 1 in 108 [2,3]. Overall, OC is the most lethal gynaecological malignancy and is recognised as the “silent killer” due to late-stage diagnosis caused by asymptomatic progression [3].

Primary OC can be categorised into non-epithelial and epithelial, germ cell, and sex cord-stromal cancer, with epithelial being the most common. Non-epithelial accounts for approximately 10% of all OC and includes mainly germ cell tumours, sex cord-stromal tumours, and some extremely rare tumours [4,5]. Germ cell tumours are extremely rare in menopausal women yet reported in the literature [6]. Ovarian carcinosarcomas, accounting for only 1–4% of all OC, are composed of an epithelial as well as a sarcomatous component [7]. Epithelial OC can be histologically categorised further into serous, clear cell, endometrioid, mucinous, or undifferentiated variety [8]. Whilst the old line of thinking hypothesised that ovarian carcinogenesis arose from metaplasia of the ovarian surface epithelium into the various subtypes (serous, mucinous, clear cell, endometrioid, and transitional), a newer accepted theory by Kurman et al. provides a dualistic model [9]. This classifies OC into Type I, which consists of clearly described precursor lesions, and Type II, where precursor lesions are not clearly described, wherein cancer may arise de novo from the tubal/ovarian epithelium. Type I, consisting of low-grade serous, mucinous, endometrioid, clear cell, and transitional cell carcinomas, is typically more indolent and presents at an earlier stage. Type II, consisting of high-grade serous carcinomas, undifferentiated carcinomas, and carcinosarcomas, behaves in a more aggressive manner, is genetically unstable, and typically presents at a later stage. Type II tumours have a high frequency of *TP53* mutations, whereas type I have specific mutations targeting cell signalling pathways, which in turn unsettle *BRCA* expression, namely *KRAS*, *BRAF*, *ERBB2*, *CTNNB1*, *PTEN*, *PIK3CA*, *ARID1A*, and *PPP2R1A* [9,10]. Serous papillary peritoneal cancer shares common molecular, histological, and clinical features with epithelial OC, mainly high-grade serous, which made it reasonable to manage the two entities similarly [11]. The recommended treatment of OC is radical surgery, followed by adjuvant chemotherapy. The therapeutic strategy of gestational OC depends on histology, stage, and gestational weeks [12]. As the proteome closely mirrors the dynamic state of cells, tissues. and organisms, proteomics has great potential to deliver clinically relevant biomarkers for OC diagnosis and treatment [13].

OC aetiology can be categorised into sporadic and hereditary. Approximately 23% of OC has a hereditary element, with the majority of those caused by defects within the BRCA DNA repair genes [14]. It is estimated that the lifetime risk of OC is 40–50% amongst *BRCA1* mutation carriers and 20–30% in *BRCA2* mutation carriers, which is significantly higher than the general population [15]. Several other genetic traits have been identified as risk factors for developing OC [14,16]. These genetic factors offer an opportunity for primary and secondary prevention strategies to reduce the risk of these high-risk individuals developing OC. Platinum compounds and poly (ADP-ribose) polymerase (PARP) inhibitors are currently the two main classes of drugs active against cancer cells, harbouring DNA damage response and repair gene alterations [17]. Genomic alterations in the DNA damage repair pathway are emerging as novel targets for treatment across different cancer types, especially OC, breast, and prostate cancer [18,19,20].

Given the lack of evidence to support screening in high-risk individuals, a preventative approach, usually in the form of surgery or chemoprevention, is the first line of management in these women [21,22,23,24,25,26,27,28,29]. Growing evidence suggests prevention strategies are both clinically and economically superior [30,31]. However, whilst most evidence applies to sporadic OC, individuals at risk of hereditary ovarian cancers (HOC) may benefit even more from targeted interventions, which may be even more cost-effective in this high-risk population [32,33].

This review paper aims to explore the aetiology and risk factors of HOC. Current preventative strategies were analysed for their clinical and economic impact to determine the most clinically and economically effective strategy for OC prevention amongst high-risk women.

## 2. Hereditary Ovarian Cancer (HOC)

HOC constitutes almost a quarter of all epithelial OC cases [34]. Family history is the strongest risk. First- and second-degree relatives with OC carry a 3.6- and 2.9-fold lifetime risk of developing OC, respectively [35].

HOC is also comprised of hereditary cancer syndromes (HCS), with mutations inherited in an autosomal dominant (AD) fashion leading to multiple primaries presenting at a young age. The two principal syndromes accounting for at least 20% of all epithelial OC are hereditary breast and ovarian cancer syndrome (HBOC) and Lynch syndrome (LS), also known as hereditary non-polyposis colorectal cancer syndrome (HNPCC) [36,37].

HBOC constitutes approximately 80% of HOC and 15% of epithelial OC cases [36]. Within HBOC, 65–85% of cases primarily stem from genomic mutations in *BRCA1* and *BRCA2* tumour-suppressor genes [38]. These encode proteins for homologous recombination (HR) to repair DNA double-strand breaks (DSB) for maintaining genomic stability. The prevalence in the UK for *BRCA1* is thought to be 0.07–0.09% and for *BRCA2* 0.14–0.22% [39]. The lifetime risk for development and average age of onset of OC from *BRCA1* and *BRCA2* is 40% and 20% and 49 to 53 years and 55 to 58 years, respectively [36,40]. More than 15% of HBOC cases arise due to mutations concerning other predisposition genes including *BARD1*, *BRIP1*, *CHEK2*, *MRE11A*, *MSH6*, *NBN*, *PALB2*, *RAD50*, *RAD51C*, or *TP53* (Figure 1). *BARD1*, *BRIP1*, *PALB2*, *RAD50, RAD51C*, *NBN*, and *MRE11A* gene mutations have been implicated in OC as part of the BRCA2/Fanconi anaemia signalling pathway in the event of nil *BRCA1* and/or *BRCA2* mutations. They constitute a significant portion of the DNA DSB repair machinery as well as next-generation sequencing (NGS)-based multigene panels [41]. The *BARD1* gene showed one novel and three previously known genomic alterations in Ratajska et al.’s cohort of 255 unselected OC cases. These were almost nil in their control group, thus highlighting their pathogenic potential [42]. The *BRIP1* mutation increases OC risk by 8-fold and decreases lifespan by almost four years [43]. The *PALB2* gene mutation is prevalent in up to 4% of BRCA-negative HBOC cases. Yang et al. found a significant association between *PALB2* pathogenic variants and OC (i.e., a relative risk of almost three) [44]. The *RAD51C* gene is prevalent in up to 2.9% of HBOC families negative for *BRCA1* and/or *BRCA2* mutations. Meindl et al. found six pathogenic variants in *RAD51C* among 1100 German families, yielding a relative risk of six for developing OC [45]. *RAD50* and *MRE11A* are constituents of the MRE11 complex. Heikkinen et al. identified germline mutations in *RAD50* and *MRE11A* among 151 HBOC families [46]. Meanwhile, Ramus et al. revealed that the prevalence of germline mutations of the *NBN* gene was very low (0.2%), hence not contributing significantly to OC risk [47]. In epithelial OC, DNA mismatch repair (MMR) deficiency is the second most common cause of HOC—only behind HR deficiency—accounting for 10–15% of HOC [48]. Furthermore, it has been reported that high mRNA levels of *MSH6*, *MLH1*, and *PMS2* were associated with a prolonged overall survival in OC. That supports the potential positive prognostic value of MMR genes in OC patients treated with platinum-based chemotherapy.

LS is the second commonest cause of HOC responsible for approximately 15% cases and 4% of epithelial OC cases [38]. It primarily involves colorectal cancer along with an increased frequency of extracolonic tumours, including endometrial, ovarian, urogenital, brain, renal, gastric, and biliary. The lifetime risk of developing OC with LS is approximately 8 to 12%, and the mean age of presentation is about 43 years. Mutations in MMR genes have been implicated in LS, namely *MSH2* (38% cases), *MLH1* (32%), *PMS2* (15%), and *MSH6* (15%) [49]. Grindedal et al. noted a 30-year OC survival of 71.5% in 144 MMR mutation carriers with OC, seemingly better than *BRCA* mutation survival [50].

Other rarer syndromes concerning HOC include Li–Fraumeni syndrome, Cowden syndrome, Peutz–Jeghers syndrome, diffuse gastric cancer syndrome, and neurofibromatosis type 1 syndrome. These arise primarily due to mutations in the *TP53*, *PTEN*, *STK11*, *CDH1*, and *NF1* genes, respectively [51]. Figure 1 illustrates the inherited gene mutations in HOC.

## 3. Prevention Strategies for Hereditary Ovarian Cancer

### 3.1. Genetic Testing and Counselling

As mentioned above, the most commonly identified genetic mutation leading to HOC is in the *BRCA* gene. Although genetic testing is widely recommended to determine the probability of inheriting a malignant condition, only ~30% of women undergo it. The concept of cascade testing involves testing an affected individual for a pathogenic hereditary variant. This, in turn, is extended to unaffected blood relatives once the specific variant is identified [52].

Genetic counselling is a multi-stage process by which individuals at risk of hereditary cancers are identified and educated regarding probability of acquisition and passing it on to the future generation. Genetic counselling is of paramount importance in HOC, as early identification of mutations can have both preventive as well as therapeutic implications, i.e., screening and/or risk-reducing interventions (RRI) ranging from chemoprevention to risk-reducing surgery (RRS) [52].

### 3.2. Screening

The concept of multimodality screening for OC in postmenopausal women among the general population gained popularity after two landmark trials—the US-based Prostate, Lung, Colon, and Ovary (PLCO) Cancer Screening Trial (1993–2001) and then the UK-based Collaborative Trial of OC Screening (UKCTOCS) (2001–2005) [53,54,55,56].

Multimodality screening incorporates a blood test, i.e., serum biomarker cancer antigen 125 (CA125), and imaging test, i.e., transvaginal ultrasound (TVS). In the PLCO, these tools were used individually [54]. However, in the UKCTOCS, they were used sequentially. Elevation in CA125 relative to baseline would imply an increase in the Risk of OC Algorithm (ROCA) score (first stage). This would in turn warrant TVS (second stage) for diagnosis [55].

OC screening has relatively poor outcomes for early detection or prevention as evident from PLCO and UKCTOCS [53,54,55,56]. The primary reason has been attributed to the generally small interval between early and advanced stage disease, largely due to its pathogenesis. The shedding of malignant cells into the abdominal cavity can cause a heightened disease progression. This short gap levies a significant challenge in OC surveillance. In addition, TVS for adnexal masses have an elevated false-positive rate, which inevitably leads to unnecessary surgical intervention. The use of CA125 as a biomarker poses challenges due the possibility of levels being raised in several benign gynaecological conditions [57]. Even in high-risk groups, there has not been an effective screening programme thus far in either reducing mortality or early detection.

Both the UKCTOCS and PLCO have demonstrated a lack of evidence for support of screening methods to significantly affect mortality [53,54,55,56]. Although evidence is generally unremarkable in the long-term, OC screening with regular pelvic examinations and multimodality screening can be used in individuals above 30 years of age. The UKCTOCS demonstrated encouraging evidence of mortality reduction by earlier detection of OC in postmenopausal women excluding those with increased risk of familial OC [58].

### 3.3. Hormonal Chemoprevention

Hormonal chemoprevention is primarily aimed at patients who have been diagnosed with a *BRCA* mutations at a younger age [59].

#### 3.3.1. Combined Oral Contraceptive Pills (COCP)

COCP usage inhibits ovulation. This reduces the number of ovulation cycles in a woman’s lifetime, thereby decreasing the overall exposure to female hormones. This in turn theoretically decreases the risk of OC, and hence, COCP have been used as a prophylactic option for OC. Narod et al. showed that patients who take COCP for any length of time carried reduced risk of OC by approximately 50%. This figure increased to 60% after 6 years of continuous COCP use [60].

A meta-analysis of 18 case-control and retrospective cohort studies (1503 OC cases) showed that the risk of OC was significantly reduced in COCP users vs. non-COCP users in *BRCA1* and/or *BRCA2* mutation carriers. They showed a 36% risk reduction in OC for every 10 years of COCP use. Importantly, this cohort did not show any association with breast cancer—this link was shown only in COCP formulations that were used prior to 1975, and more recent formulations have not shown a significant association [61].

The largest pooled data on COCP users demonstrated a prevention of approximately 200,000 diagnoses of OC worldwide, leading to a prevention of 100,000 deaths. Large meta-analyses and systemic reviews have shown a strong benefit of COCP use, demonstrating an odds ratio (OR) of 0.73 (95% confidence interval (CI) 0.66–0.81) and a lifetime reduction risk of 0.54%. These pooled datasets, however, have not differentiated between BRCA-positive and non-BRCA patients [62].

The clear benefit of COCP needs to be weighed up against theoretical risk of breast cancer, as they have shown to mildly elevate the risk of breast cancer in the general population. However, studies have proven inconclusive on this link for *BRCA* mutation carriers [63,64]. Based on all data, BRCA patients should be counselled on these theoretical risks as well as benefits of COCP use and be advised caution when using this method as a preventative measure for OC.

#### 3.3.2. Other Hormonal Agents

Progestin-only pills (POP), etonogestrel subdermal implants, and injectable depot medroxyprogesterone acetate (DMPA) may have a protective role against OC. Progesterone increases the expression of the tumour suppressor gene *P53* and has been shown to possess anti-proliferative, anti-metastatic properties against OC cells in vitro. These findings have not been translated to clinical trials yet. A prospective nationwide cohort study indicated that the use of progestogen only products did not significantly reduce the risk of OC. No large studies have been conducted on the effects of POP in high-risk BRCA positive individuals thus far; therefore, it remains to be seen whether this intervention may be used in the future to protect BRCA carriers from ovarian cancer [65].

### 3.4. Surgical Prevention

#### 3.4.1. Bilateral Tubal Ligation (BTL)

BTL for sterilisation has been shown to reduce risk of OC. However, there are limited studies on applying BTL to BRCA carriers. In the BRCA1 population with previous BTL, Narod et al. in 2001 demonstrated a 39% overall risk reduction in developing OC. This decreased to a 28% risk reduction with concomitant COCP usage. These risk-reducing effects of BTL were not reproducible for BRCA2 carriers [66]. A 2011 meta-analysis showed women with previous BTL had a 34% overall risk reduction in developing OC, but there was no significant reduction in those with borderline or mucinous tumours. This protective effect was maintained at 14 years after surgery [67].

The aforementioned protective effect can be due to a mechanical barrier blocking the retrograde flow of carcinogens from the vagina or perineum. More specifically, epithelial cells embryologically derived from the Mullerian ductal system including FT and endometrium can predispose to endometrioid or serous variants of OC. Prevention of their ascent courtesy BTL are in sync with Cibula et al.’s results [67].

In their meta-analysis, Rice et al. showed the relative risk amongst BRCA patients with vs. without BTL was not statistically different amongst the general population (risk ratio (RR) 0.64 vs. 0.7). Although results show promise, long-term implications of the procedure have not been well-documented, especially in BRCA-positive patients [68].

#### 3.4.2. Risk-Reduction Bilateral Salpingo-Oophorectomy (RRBSO)

RRBSO is considered the gold-standard treatment in BRCA positive patients for prevention of OC. This is based on the traditional hypothesis that the ovarian surface epithelium inclusions occurring during ovulation are subject to cellular metaplasia, leading to various subtypes of OC. OC once initiated would then continue disseminating via the FT to the other gynaecological organs and peritoneal cavities. A prospective study by Kauff et al. have shown 70–85% reduction in OC and overall significant reduction in mortality [69]. Rebbeck et al. showed similar results but had a longer follow up period—they showed a 96% reduction in BRCA-related gynaecological cancer [70].

The optimal age for RRBSO of 35 to 40 years is primarily based on the positive outcomes of the aforementioned studies as well as the relatively increased risk of developing BRCA1-related OC after 40 years. Interestingly, RRBSO can be delayed in BRCA2-related OC cases, as their relative risk of malignancy does not start until aged 50. The rate of OC is low under 40 but reaches approximately 10% by age 50 in BRCA 1 carriers. In BRCA 2 carriers, this remains low until aged 50 [69,70].

Despite RRBSO being the standard RRS, it is not without its risks. The procedure induces a surgical menopause and oestrogen lack on average 10 years earlier than those who have their ovaries intact [71]. Long-term follow up studies involving general population have been linked with increased risk of osteoporosis, stroke, cardiovascular disease, and neurocognitive decline [72]. However, similar studies conducted in the BRCA population show a paucity of data other than menopausal symptoms (reduced libido, vaginal dryness, dyspareunia), which may not even be fully relieved by COCP use post RRBSO [73].

#### 3.4.3. Risk-Reducing Bilateral Salpingectomy (RRBS)

One of the more popular theories of epithelial OC tumour precursors originating in the 1990s has an extra-ovarian origin, namely from the fimbrial end of the fallopian tube epithelium, which constitutes the tubo-ovarian complex. These are primarily serous tubal intraepithelial carcinomas (STIC). Labidi-Galy et al. performed whole-genome sequencing and copy number analysis of laser capture micro dissected fallopian tube lesions in BRCA-related OC patients post RRBSO. The majority of tumour-specific alterations were prevalent in STIC, and a window of 7 years between development of STIC to epithelial OC was noted [74].

Based on this theory, Faulkner et al. showed a 50% risk reduction for OC in favour of RRBS alone [75]. RRBS is a less radical option than RRBSO for younger women not keen on oophorectomy in fear of early surgical menopause. However, it might still be in its early days as a standard RRS in HBOC owing to its doubtful efficacy and risk reduction of breast cancer. This is due to the lack of protective effect against breast cancer conferred by oophorectomy [76].

#### 3.4.4. Hysterectomy

Hysterectomy is slowly emerging as a RRS to prevent OC. According to Rice et al., the protective effect can be due to prevention of retrograde flow as mentioned prior. Another theory can be the “screening effect”, i.e., surgeons removing possible pre-malignant lesions on direct visualization. Cutting off blood supply to the ovary would decrease the oestrogen production and hence reduce overall hormone exposure, implying another protective mechanism [68].

A Canadian population-based retrospective cohort compared outcomes of 43,931 low-risk women from 2008–2011 that underwent hysterectomy with and without RRBSO/RRBS/BTL. With regards to surgical outcomes, these hysterectomy-based approaches proved to be feasible, safe, efficacious, and had minimal complications [77]. However—as an RRS—more research is needed to determine the effect of hysterectomy in terms of OC risk reduction in BRCA carriers as an opportunistic or a stand-alone procedure.

## 4. International Guidelines

A wide range of practice guidelines, protocols, and recommendations regarding genetic testing have been formulated by various professional organisations for the high-risk OC population who would benefit from genetic counselling and preventive strategies. Select guidelines are covered in this section and are depicted in Table 1.

### 4.1. Society of Gynaecologic Oncology (SGO)

The SGO primarily looked at HBOC and LS. They recommended genetic testing and thereafter genetic counselling for all women at increased predisposition for OC due to personal or family history. These included cases with high-grade epithelial OC/tubal/peritoneal cancer; breast cancer ≤ 45 years/≤ 50 years with a limited family history; breast cancer with one first-degree relative having breast cancer ≤5; 0 years or with epithelial OC/tubal/peritoneal cancer at any age; breast cancer with ≥2 first-degree relatives having breast cancer at any age or pancreatic/aggressive prostate cancer; triple-negative breast cancer ≤ 60 years; breast cancer and Ashkenazi Jewish (AJ) ancestry; and two breast primaries with first one diagnosed before 50 years. Also included in this cohort are women unaffected with cancer but with first-degree relative diagnosed with any of the aforesaid factors or with male breast cancer or with known *BRCA* mutation [21].

The SGO also recommends genetic assessment for women with a high likelihood for LS. They include ones with endometrial or colorectal cancer with evidence of microsatellite instability or loss of dMMR protein (MLH1, MSH2, MSH6, PMS2); first-degree relative affected with endometrial or colorectal cancer either diagnosed before 60 years or at risk for LS; first- or second-degree relative with a known MMR gene mutation [21]. Technically, immunohistochemical testing of the MMR machinery may give different results for a given germline mutation, and it has been suggested that this may be due to somatic mutations [78].

The SGO did not yield any evidence to support screening. They advocated usage of COCP in HBOC. RRBSO was recommended between 35 and 40 years as a standard of care. RRBS was to be considered if RRBSO was declined although it would not act as a substitute due to offering only OC and nil breast cancer risk reduction [22].

### 4.2. European Society of Medical Oncology (ESMO)

The main area of interest for ESMO was HBOC. They recommended genetic testing for all women above 25 years of age hailing from families harbouring pathogenic *BRCA1* and/or *BRCA2* variants. If positive, then genetic counselling with regard to screening and RRI options was deemed mandatory. In the event of individuals declining testing or counselling, the same screening recommendations as for BRCA mutant carriers are to be followed. All BRCA carriers should be encouraged to attend high-risk follow up clinics. The aforesaid were level evidence V and grade recommendations B (V B) [23].

The ESMO considered COCP to be a risk-reducing measure (II C). They advocated 6-monthly multimodality screening from the age of 30 years as a screening measure (V C). Similar to SGO recommendations, the ESMO also advocated RRBSO from age 35 to 40 years (II B), as it was the most effective RRI for OC (IA). They did not recommend RRBS alone outside the clinical trial setting (V C) [23].

### 4.3. American College of Obstetrics and Gynaecologists (ACOG)

The ACOG published guidelines on both HBOC and LS. On the basis of consistent scientific evidence, they recommended genetic counselling for all women with epithelial OC and those with personal or family history of breast cancer or OC (level B). On the basis of consensus, they recommended genetic testing in women where detailed risk assessment revealed a high probability of an inherited cancer syndrome. The recommended testing modalities of HBOC were *BRCA* mutation and multigene panel testing (level C). According to the ACOG, RRBSO is the sole intervention to reduce OC-specific mortality and should be implemented in BRCA1 carriers aged 35 to 40 years and BRCA2 carriers aged 40 to 45 years. They used TVS or serum CA125 as tools more for short-term surveillance rather than screening at 30 to 35 years of age before they could have RRBSO (level C) [24].

With regard to LS, they recommended genetic testing in all women with endometrial or colorectal cancer irrespective of age of diagnosis and also those with a positive family history (level B). For unaffected females having a first-degree relative with endometrial or colorectal cancer diagnosed before 60 years or those with positive personal/family history, genetic risk assessment was advised (level B). They also advocated RRBSO and total hysterectomy as a potent risk-reducing modality for the aforesaid women in their 40s (level B) [25].

### 4.4. Manchester International Consensus Group (MICG)

Meanwhile, the MICG concentrated on LS. They recommended screening for LS in OC patients who were diagnosed ≤50 years or at any age if having non-serous and non-mucinous histology (level C). Regarding methodology of screening, they strongly recommended NGS, which would include *BRCA* and LS genes (level B/C). In the event of suspicious LS with no proven LS-associated pathogenic variant on NGS, the MICG recommended searching for association with other somatic or germline variants (level B) [26].

Akin to the guidance mentioned aforesaid, the MICG strongly recommended that RRBSO and risk-reducing hysterectomy (RRH) should be offered in MMR pathogenic variants at 35 to 40 years of age following completion of family/childbearing (level B). Additionally, women undergoing the above are to be offered oestrogen-only hormone replacement therapy (HRT) until natural menopause (level B) [26].

With regard to chemoprevention, they passed a level B recommendation for COCP usage to reduce risk of OC and endometrial cancer and a level A recommendation for aspirin to reduce risk of colorectal cancer in all MMR variants [26].

### 4.5. E. United States Preventive Services Task Force (USPSTF)

HBOC was the focus for the USPSTF. They recommended familial risk assessment and,, accordingly genetic counselling and/or testing for women with a personal or family history of breast/ovarian/tubal/peritoneal cancer or pedigree with *BRCA* mutations (level B). The Ontario Family History Assessment Tool, the Manchester Scoring System, the Referral Screening Tool, and the Pedigree Assessment Tool are some examples of risk assessment tools used. They recommended against risk assessment/counselling/testing in women without personal or family history of the aforesaid (level D) [27].

### 4.6. American Society of Clinical Oncology (ASCO)

The ASCO also focused on HBOC. They recommended germline *BRCA1* and/or *BRCA2* and other susceptibility testing for all patients diagnosed with epithelial OC irrespective of personal or family history. If found negative on germline testing, then somatic *BRCA1* and/or *BRCA2* or likely pathogenic variant testing is advocated. First- or second-degree relatives of aforesaid women with positive germline mutation are to be offered risk assessment, genetic counselling, and testing. Strength of the two aforesaid recommendations were strong. They also recommend offering somatic testing for MMR to women diagnosed with any histologic variant of epithelial OC, with strength being moderate [28].

### 4.7. National Comprehensive Cancer Network (NCCN)

The primary interest of the NCCN was HBOC and Li–Fraumeni syndrome. Testing for *BRCA1*, *BRCA2*, *CDH1*, *PALB2*, *PTEN*, and *TP53* was recommended. In terms of HBOC, testing criteria included individuals with positive personal history of cancer, i.e., epithelial OC/exocrine pancreatic cancer at any age or breast cancer with the following clauses. They include breast cancer diagnosed at any age with a AJ ancestry/≥1 first-/second-degree relative with breast cancer < 50 years or OC; pancreatic or prostate cancer at any age/≥3 cases of breast cancer, including individual and close relatives; breast cancer diagnosed at age ≤ 45 years or ≤60 with triple-negative breast cancer; breast cancer diagnosed at age 46 to 50 years with unknown family history; second breast cancer diagnosed at any age or ≥1 close relative with breast/OC/pancreatic/prostate cancer at any age. With regard to Li–Fraumeni syndrome, testing was recommended for individuals with pathogenic *TP53* variant or who met classic Li–Fraumeni syndrome or Chompret criteria [29].

If criteria for HBOC were met, and testing was positive for pathogenic variants, genetic counselling for RRI primarily RRSO was mandatory. The NCCN recommended RRBSO at 35–40 years in *BRCA1* variants and 40–45 years in *BRCA2* counterparts upon family completion. In the event of RRBSO being declined, multimodality screening commencing at 30 to 35 years of age warranted consideration [29].

They recognised the OC risk reduction due to COCP usage but noted conflicting breast cancer risk data. Hence, they expressed the need for large prospective trials to identify the conflicting relationship before passing a strong recommendation [29].

## 5. Comparative Analysis of Cost-Effective Prevention Strategies

The cost of treatment per patient with OC remains the highest among all cancer types. As an example, the average initial cost in the first year can amount to around USD 80,000, whereas the final year cost may increase to USD 100,000 [79]. Over the last decade, cost-effective strategies for early detection and prevention of OC have been investigated. Here, we discuss a comparative analysis among two subgroups of women—the general population and women with HOC risk (Table 2).

### 5.1. General Population

Based on data from the PLCO Trial, Drescher et al. found that annual screening using rising CA125 to predict candidates for TVS yields a mortality reduction of 13% and incremental cost-effectiveness ratio (ICER) of USD 89,000 per year of life saved (YLS). This was a better proposal than semi-annual screening with a higher mortality reduction (20%) but with lower cost effectiveness, i.e., ICER of USD 117,000/YLS [80].

Kearns et al. analysed the cost effectiveness of sequential ROCA based multimodal screening in the UK setting, based on the UKCTOCS Trial. Compared to TVS alone, multimodal screening was more effective and less expensive. Compared to no screening, multimodal screening was of course more effective and expensive, generating an ICER of almost GBP 9000 per quality-adjusted life years (QALY) [81].

Menon et al. performed a within-trial (UKCTOCS) economic evaluation of multimodal screening vs. no screening. Keeping the CA125-ROCA unit cost of GBP 20 per patient, multimodal screening had an ICER of around GBP 90,000 per life years gained (LYG) compared to no screening. This figure down trended to GBP 78,000 when the unit cost was lowered to GBP 15 per patient [82]. Moss et al. established a cost-effectiveness model of OC in the USA using data extrapolated from UKCTOCS. Compared to no screening, multimodal screening for postmenopausal women starting at age 50 reduced mortality by 15%, with an ICER of around USD 105,000 to USD 155,000 [83].

The aforesaid health economic analyses therefore yielded similar results [81,82,83]. Contrary to the above—using UKCTOCS data—Naumann et al.’s health economic model predicted that the ROCA-based multimodal screening can reduce overall OC mortality, but costs are substantiative. More specifically, commencing screening at age 50 for 20 years yielded an absolute decrease in mortality of 6% at a cost of around USD 590,000 per LYG. The same for 30 years saw absolute decrease of 9% costing at approximately USD 760,000. Thus, costs would have to be reduced at least ten-fold for a marginal decrease in mortality rates for this screening to be affordable [84].

As discussed previously, most of the high-grade serous OC are noted to originate from the fallopian tube [74,85]. This fact found its implications in opportunistic salpingectomy, whereby uterine tubes can be permanently removed at any surgical opportunity. The procedure can be salpingectomy with hysterectomy or salpingectomy instead of tubal ligation [86]. Kwon et al. used health economic models to analyse both the aforesaid in the Canadian population. It was noted that salpingectomy with hysterectomy was less expensive than hysterectomy or bilateral salpingo-oophorectomy (BSO) alone, as the costs were approximately USD 11,000, USD 11,200, and USD 12,600, respectively. It also reduced OC risk by 38% when compared to hysterectomy alone. Moreover, salpingectomy was more expensive than tubal ligation, as their costs were around USD 9700 and USD 9300, respectively. However, it had an ICER of USD 27,000 per LYG. It had an OC reduction rate of 29% compared to sterilisation [86].

**Table 2 ijerph-19-12057-t002:** Cost effectiveness of prevention strategies.

Population		Year	Reference	Risk-Reducing Strategy		Cost Effectiveness of Intervention (ICER)	Comments
General		2012	[80]	Annual screening CA125 to predict candidates for TVS		USD 89,000/YLS, 13% mortality reduction	
				Semi-annual screening CA125 to predict candidates for TVS		USD 117,000/YLS, 20% reduction in mortality	
		2016	[81]	MMS—Sequential ROCA		GBP 9000/QALY	
		2017	[82]	MMS with per unit cost of GBP 20 per patient		GBP 90,000/LYG	
				MMS with per unit cost of GBP 15 per patient		GBP 78,000/LYG	
		2018	[83]	MMS vs. no screening		15% mortality reduction, ICER: USD 105,000–155,000	
		2018	[84]	ROCA-based MMS commenced at age 50 for 20 years		6% decrease in mortality with USD 590,000/LYG	
				ROCA-based MMS commenced at age 50 for 30 years		9% decrease in mortality with USD 760,000/LYG	
		2015	[86]	RRBM		USD 27,000/LYG	
Hereditary Ovarian Cancer	BRCA 1/2	1998	[87]	RRBO at 30 years of age		2.6 years survival improvement and QALY of 0.5 in favour of PO	
		2006	[88]	RRBSO at age 35			Most cost-effective with quality adjustment
		2008	[89]	RRBSO			85% decrease in BRCA1 OC, no statistically significant effect on BRCA2 OC
		2008	[90]	RRBSO + RRBM		EUR 496/LYG	
				RRBSO alone		EUR 1284/LYG	
		2011	[91]	RRBM vs. RRBSO vs. RRBM+RRBSO vs. chemoprevention vs. surveillance	BRCA1	PBSO—USD 1741/QALY	
					BRCA2	PBSO—USD 4587/QALY	
		2013	[92]	RRBS vs. RRSDO vs. RRBSO	BRCA1	PSDO—USD 37,800/QALY	RRBSO yielded highest risk reduction, life expectancy and lowest cost, RRSDO had highest ICER
					BRCA2	PSDO—USD 89,700/QALY	
		2018	[93]	RRBM vs. RRBSO vs. RRBM+RRBSO at age 30 vs. RRBM+RRBSO at age 40		PBM + PBSO at age 30—cost of EUR 29,000 and 17.7 QALY gained or 19.9 LYG	
	Lynch Syndrome	2008	[94]	Annual screening from age 30 followed by RRH + RRBSO at age 40 vs. Only screening from age 30 vs. Only RRH +RRBSO at age 40 or 30 vs. No intervention		Annual screening from age 30 followed by PH + PBSO at age 40—USD 195,000/QALY	
		2011	[95]	RRBSO+RRH at age 30		USD 23,400 per patient and QALY-26	
	Strong Family History	2019	[96]	No mutation testing vs. Cascade testing followed by RRS		Cascade testing followed by RRS—USD 9000–10,000 per QALY	
		2019	[97]	Intensified surveillance followed RRS (RRBM/ RRBSO/ RRBM+RRBSO)		EUR 17,000/QALY and EUR 22,000/LYG	Prevented one-third of malignancies, RRBM + RRBSO was the most cost-effective RRS
	Ashkenazi Jewish Women	1999	[98]	Surveillance followed RRS (RRBM + RRBSO)		USD 21,000/LYG	
		2009	[99]	Screening + RRBSO vs. no screening		USD 8300/QALY	
		2014	[100]	Population-based screening vs. Family-based screening, both followed by RRS		Population-based screening—GBP 2079/QALY	Done in population with index cases of 4 AJ Grandparents
		2017	[101]	Population-based screening followed by RRBSO	1 AJ Grandparent	GBP 2793/USD 7110/QALY (UK/US)	Highly cost-effective even in varying AJ ancestry
					2 AJ Grandparents	GBP 301/USD 7366/QALY (UK/US)	
					3 AJ Grandparents	GBP 1759/USD 14,032/QALY (UK/US)	
					4 AJ Grandparents	GBP 2589/USD 17,786/QALY (UK/US)	
	Sephardi Jewish Women	2018	[102]	Population based screening vs. Family based screening		£67/$308/QALY (UK/US)	

AJ, Ashkenazi Jews; LYG, life years gained; ICER, incremental cost-effectiveness ratio; MMS, multimodality screening; QALY, quality adjusted life years; ROCA, Risk of Ovarian Cancer Algorithm; RRBM, risk-reducing bilateral mastectomy; RRBO, risk-reducing oophorectomy; RRBSO, risk-reducing bilateral salpingo-oophorectomy; PBM, prophylactic bilateral mastectomy; PBSO, prophylactic bilateral salpingo-oophorectomy; RRH, risk-reducing hysterectomy; RRS, risk-reducing surgery; RRSDO, risk-reducing bilateral salpingectomy with delayed oophorectomy; TVS, transvaginal scan; YLS, years of life saved.

### 5.2. Women with HOC Risk

Women with HOC risk comprise of BRCA mutation carriers, LS carriers, those with a strong family history or high individual familial risk, and those belonging to certain Jewish populations.

### 5.3. BRCA Mutation Carriers

As discussed above, BRCA mutation carriers have a significant lifetime risk of OC and breast cancer. Therefore, RRS as a preventive strategy has been deemed beneficial. Grann et al. offered RRS including risk-reducing bilateral oophorectomy (RRBO)/risk-reducing bilateral mastectomy (RRBM) to their cohort of 30-year-old BRCA carriers, whom they followed-up for 50 years. Compared to surveillance, RRBO improved survival by 0.4 and 2.6 years in the low-risk and high-risk models, respectively. The high-risk model also yielded QALY of 0.5 in favour of RRBO [87]. Anderson et al. used a similar cohort 35 to 50 years of age where RRS, including RRBSO, was offered at 35. RRBSO was by far the most cost-effective strategy for mutation carriers with quality adjustment [88]. Kauff et al.’s prospective, multicentre study followed-up around 1000 30-year-old BRCA carriers for 3 years who chose RRBSO vs. surveillance. It was noted that RRBSO yielded 85% decrease in BRCA1-associated OC risk but no statistically significant effect on BRCA2-associated OC [89]. A Norwegian study led by Norum et al. on BRCA1 carriers compared RRBSO at 35 years (with or without RRBM at 30 years) with no RRS. RRBSO with and without RRBM underwent 6.4 and 3.1 discounted LYG, respectively. Median survival improved by 25 and 16 years, respectively. The ICER for RRBSO alone and RRBSO + RRBM was EUR 1284 and EUR 496 per LYG respectively, hence concluding that RRBSO with or without RRBM was cost-effective [90]. Grann et al. performed a comparative effectiveness study on a cohort of BRCA carriers aged 30 to 65 years. Among different RRI, such as RRBM, RRBSO, RRBM + RRBSO, chemoprevention, and surveillance, the most dominant intervention was RRBSO alone. It was also the most cost-effective for BRCA1 (ICER USD 1741/QALY) as well as BRCA2 carriers (ICER USD 4587/QALY) [91]. However, RRBSO has significant adverse effects such as premature menopause as mentioned earlier. Hence Kwon et al. compared outcomes of risk-reducing salpingectomy with delayed oophorectomy with RRBSO and RRBS. Although RRBSO yielded the highest risk reduction, life expectancy, and lowest cost, risk-reducing salpingectomy with delayed oophorectomy had the highest ICERs for *BRCA1* and/or *BRCA2* (USD 37,800/89,700 per QALY). It thus proved a considerable alternative to avoid the adverse effects of RRBSO [92]. A German experience developed by Muller et al. on around 5900 BRCA carriers compared RRBM, RRBSO, AND RRBM + RRBSO all at age 30 and RRBM + RRBSO at age 40. They noted that with a cost of around UER 29,000 and gain in QALY 17.7/LYG 19.9, RRBM + RRBSO at 30 years was the most cost-effective strategy, with RRBM + RRBSO at 40 years being a close second. Hence, RRBM + RRBSO was their choice of RRS [93].

### 5.4. LS Carriers

As discussed previously, women with LS/HNPCC are at a lifetime risk of OC and endometrial cancer. Taking this into consideration, screening and RRS, i.e., RRH + RRBSO, have been recommended as prevention strategies. Kwon et al. compared five RRI in their cohort of LS carriers, namely combined strategy (annual screening from 30 years and RRH + RRBSO at 40 years), screening only from 30 years, RRH + RRBSO at 40 years, and 30 years and no RRI. The combined strategy was found to be the most cost-effective with an ICER of USD 195,000 per QALY [94]. In another model, Yang et al. compared RRH + RRBSO at age 30 with surveillance (annual gynaecological screening or exam). They noted that RRH + RRBSO was the most dominant strategy, as it had the lowest cost per patient (USD 23,400) and the highest QALY (26) [95].

### 5.5. Women with Strong Family History

We earlier discussed about family history of cancer being the strongest predisposing factor and raised the importance of genetic testing and genetic counselling concerning *BRCA1* and/or *BRCA2* mutation. Knowledge of this paves the way for planning preventive strategies and estimate absolute risk reduction [96,97]. Hoskins et al. used Canadian epithelial OC patient data and implemented no mutation testing (treatment if cancer developed) vs. cascade testing (if index patient tested BRCA-positive, then first- and then second-degree relative would forego testing as a cascade effect) followed by RRBSO if positive. Overall, 390 out of 2786 index cases, 366 out of 766 first-degree relatives, and 49 out of 207 second-degree relatives tested BRCA-positive and in turn had RRS. Fifty-nine epithelial OC were prevented successfully. The budget impact for cascade testing/RRS was almost USD 7,000,000 as compared to USD 10,000,000 for treatment in case of no testing, hence bringing about a cost saving of almost USD 3,000,000. The ICER was around USD 9000 or USD 100,000 per QALY [96]. A similar approach was executed by the German Consortium for HBOC (GC-HBOC). Based on individual familial cancer risk, genetic testing was offered to index patients with BRCA mutational probability of ≥10%. If positive, options would be intensified surveillance or RRS. Muller et al. used this German data for analysing a cohort consisting of 35-year-old women. In total, 1540 out of 4380 were BRCA carriers, and 93% of them chose RRS along the lines of RRBM, RRBSO, and RRBM + RRBSO. Compared to the no-test strategy, their screen and test strategy established and ICER of approximately EUR 17,000 per QALY and EUR 22,000 per LYG. This successfully prevented one-third of malignancies and thus caused 20% mortality reduction. RRBM + RRBSO proved to be the most cost-effective RRS modality [97].

### 5.6. Ashkenazi Jewish (AJ) Women

The AJ population has an extremely significant *BRCA* mutational burden of 2.5%. Akin to the Muller et al. strategy, Grann et al.’s screen and RRBM + RRBSO group turned out to be the most cost-effective among the other RRS. The ICER and average survival when compared to no screen was around USD 21,000 per LYG and 38 days, respectively [98]. Rubenstein et al. conducted a decision analysis on a population-based genetic screening program (BRCA) for the American AJ population of 6.4 million. It was especially aimed at women from 35 to 55 years of age, so their working cohort was approximately 0.96 million. Their model compared screening and RRBSO to no screening and reduced OC incidence by 0.29% or 2800 cases. With regard to life expectancy, they noted a prolongation in average survival in terms of 0.0369 LYG (13 days) or 0.0459 QALY (17 days). The ICER was USD 8300 per QALY [99]. A similar model was implemented by Manchanda et al. on the British AJ population. The cohort consisted of women of age ≥ 30 years and a *BRCA* mutational burden of ≥10%, thus amounting to 0.11 million. They were offered RRS in terms of RRBSO or RRBM. Population-based genetic screening was compared to family-history-based screening. It reduced OC incidence by 0.34% or 276 cases. Average survival was prolonged by 33 days (0.1 QALY or 0.09 LYG). The ICER was GBP 2079 per QALY [100]. Interesting to note, this model was conducted on index cases with 4 AJ grandparents. Therefore, Manchanda et al. went on to further develop their previous model on 1–4 AJ grandparents. This time, they performed it on both the British and American AJ population. Life expectancy gain among UK/USA women was noted to be 15/12, 22/17, 28/22, and 33/26 days for one, two, three, and four AJ grandparents, respectively. The ICER for screening + BSO are GBP 2589/USD 17,786, GBP 1759/USD 14,032, GBP 301/USD 7366, and GBP 2793/USD 7110 per QALY for UK/USA women with four, three, two, and one AJ grandparent, respectively. Hence, they proved that their model was highly cost-effective in varying AJ ancestry [101].

### 5.7. Sephardi Jewish (SJ) Women

The SJ population, with an overall *BRCA* prevalence of 0.7% and presence of founder mutations from the AJ population, cannot be ignored [102]. Referencing Manchanda et al.’s model [101], Patel et al. included British and American SJ women ≥ 30 years of age for their cohort. Compared to family-history-based testing, population genetic testing yielded average life expectancy gain by 12 months in both UK and US populations. ICER was at GBP 67 per QALY in the UK and USD 308 per QALY in the USA. This study concluded the cost effectiveness of population-based *BRCA* testing in Jews irrespective of Ashkenazi/Sephardi descent [102].

## 6. Conclusions

OC remains a deadly cancer with poor prognosis. Public health strategies for OC prevention and early treatment are paramount to improve disease outcomes. There are still no effective tools for general population screening. This is also reflected economically, with mixed results from several studies analysing the cost effectiveness of screening programmes. Improvements in genetic testing have aided the identification of women at higher risk of developing OC. There is a lack of clear consensus in guidelines on the optimum time and age for prophylactic surgery or genetic counselling services. Nevertheless, it is clear that prophylaxis is effective at reducing OC incidence and mortality. Furthermore, the literature suggests a clear economic benefit in prophylactic measures within high-risk groups. Surgical prophylaxis, namely BSO, constitutes the main management option, boasting the most favourable outcomes and superior cost effectiveness. However, this comes with the risks and long-term complications of the procedure, which is an important consideration given that the literature suggests increased clinical benefit and cost effectiveness when carried out earlier in life. Whilst chemoprevention may be an alternative to young women who wish to preserve their fertility, there is currently limited evidence on its economic impact. Given the complexity and uniqueness of each patient, including associated risks for surgical complications, a personalised management is mandated through multidisciplinary team discussion that takes into consideration the fertility, mental wellbeing, endocrine health, and gynaecologic health of each patient.

## Figures and Tables

**Figure 1 ijerph-19-12057-f001:**
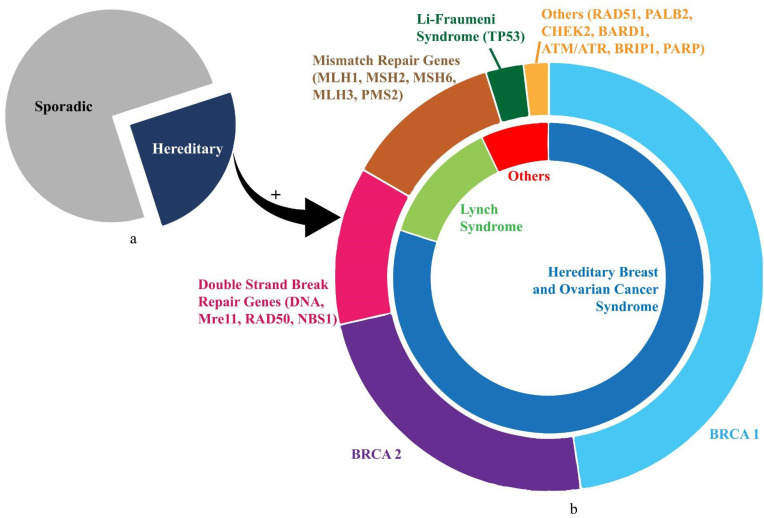
The illustration shows distribution of gene mutations in ovarian cancer. (**a**) Proportion of sporadic (75%) and hereditary (25%) ovarian cancer cases. (**b**) Details of HOC. The inner circle shows the proportions contributed by HBOC and Lynch Syndrome. The outer circle demonstrates the divisions shared by prominent genetic mutations (*BRCA1*, *BRCA2*, *TP53*) and mutation groups (mismatch repair genes, double-strand-break repair genes) corresponding approximately to the syndromic association in the inner circle.

**Table 1 ijerph-19-12057-t001:** Preventive strategies recommended by various international organisations for hereditary ovarian cancer syndromes.

Organization	Year	Population	Recommendation			References
			Screening	Risk-Reducing Intervention		
				Chemoprevention	Surgery	
Society of Gynaecologic Oncology	2015	1. HBOC 2. LS	-	Long-term COCP for HBOC	HBOC—RRBSO at 35 to 40 years of age; RRBS if RRBSO declined	[21,22]
European Society of Medical Oncology	2016	HBOC	6 monthly MMS commencing from 30 years of age	Long-term COCP	RRBSO at 35 to 40 years of age	[23]
American College of Obstetrics and Gynaecologists	1. 2017, 2. 2014	1. HBOC 2. LS	MMS as short-term surveillance (not screening) in HBOC at 30 to 35 years prior to RRBSO	-	1. HBOC—RRBSO in *BRCA1* variant 35–40 years and *BRCA2* variant 40 to 45 years 2. LS—RRBSO + RRH around 40 years	[24,25]
Manchester International Consensus Group	2019	LS	Multigene panels using NGS technology involving *BRCA* and LS-susceptible genes	Long-term COCP	RRBSO + RRH at 35–40 years following childbearing	[26]
United States Preventive Services Task Force	2019	HBOC	Familial risk assessment screening	-	-	[27]
American Society of Clinical Oncology	2020	HBOC	Germline GT for all women diagnosed with EOC	-	-	[28]
National Comprehensive Cancer Network	2021	1. HBOC 2. LFS	MMS in HBOC at 30 to 35 years if RRBSO declined	-	HBOC-RRBSO in *BRCA1* variant 35–40 years and *BRCA2* variant 40 to 45 years	[29]

COCP, combined oral contraceptive pill; EOC, epithelial ovarian cancer; GT, genetic testing; HBOC, hereditary breast and ovarian cancer; LS, Lynch syndrome; LFS, Li–Fraumeni syndrome; MMS, multimodal screening; NGS, next-generation sequencing; RRBS, risk-reducing bilateral salpingectomy; RRBSO, risk-reducing bilateral salpingo-oophorectomy; RRH, risk-reducing hysterectomy.

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
