# Peer review of "Hereditary Ovarian Cancer: Towards a Cost-Effective Prevention Strategy"

_ijerph, 2022, doi:10.3390/ijerph191912057_

Round 1
Reviewer 1 Report
In this review entitled “ Hereditary Ovarian Cancer: Towards a Cost-Effective Prevention Strategy”, Ghose et al address a very important topic, very relevant to the focus of the journal and special issue. The structure of the draft is clear and exhaustive, maybe in several points a little too verbose, and it is well written.
Overall, in my opinion the original draft should be accepted for publication, and only some minor revisions are requested.
Specific comments:
- Authors should improve resolution of Figure 1.
- By adding more specific details about the mechanism of protection from ovarian cancer carried out by COPC therapy could help readers to understand.
- By adding more specific details about the surgical prevention procedures could help readers to understand.
- In Table 1 I don’t understand because the sections dedicated to ASCO and USPTF are empty, given that in the text some guidelines are reported for these associations.
- Please make explicit AJ acronym in title of page 13.
- Maybe some additional comments could be added to conclusions section about the advantage and disadvantage in economic terms of the several prevention strategies, fully discussed towards the review.
Author Response
Dear Editor and Reviewers,
I am pleased to resubmit for publication the revised version of the manuscript ijerph-1926529, entitled “Hereditary Ovarian Cancer: Towards a Cost-Effective Prevention Strategy”.
Thankfully the reviewers provided us with a great deal of guidance, regarding how to better position the article. We are hopeful you agree that this revision will update our comprehensive review. All the comments have been addressed, as shown in the revised version of the manuscript, along with this point-by-point response to the reviewers' comments.
All corresponding are blue changes in the manuscript.
Reviewer #1:
In this review entitled “Hereditary Ovarian Cancer: Towards a Cost-Effective Prevention Strategy”, Ghose et al address a very important topic, very relevant to the focus of the journal and special issue. The structure of the draft is clear and exhaustive, maybe in several points a little too verbose, and it is well written.
Overall, in my opinion the original draft should be accepted for publication, and only some minor revisions are requested.
Specific comments:
- Authors should improve resolution of Figure 1.
- By adding more specific details about the mechanism of protection from ovarian cancer carried out by COPC therapy could help readers to understand.
- By adding more specific details about the surgical prevention procedures could help readers to understand.
- In Table 1 I don’t understand because the sections dedicated to ASCO and USPTF are empty, given that in the text some guidelines are reported for these associations.
- Please make explicit AJ acronym in title of page 13.
- Maybe some additional comments could be added to conclusions section about the advantage and disadvantage in economic terms of the several prevention strategies, fully discussed towards the review.
Response:
Thank you for your comments.
-
-
Figure 1 changes have been made.
-
Details added for protective mechanism of COCP added to main text in concerned section -> COCP usage inhibits ovulation. This reduces the number of ovulation cycles in a woman’s lifetime, thereby decreasing the overall exposure to female hormones. Such in turn theoretically decreases the risk of OC and hence COCP have been used as a prophylactic option for OC.
-
Details regarding protective mechanism of surgical procedures are all there now. For RRBS it was mentioned so no changes in that section. For BTL have added text -> The aforementioned protective effect can be due to a mechanical barrier blocking the retrograde flow of carcinogens from the vagina or perineum. More specifically, epithelial cells embryologically derived from the Mullerian ductal system including FT and endometrium can predispose to endometrioid or serous variants of OC. Prevention of their ascent courtesy BTL are in sync with Cibula et al’s results [67]. For RRBSO have added text -> This is based on the traditional hypothesis that the ovarian surface epithelium inclusions occurring during ovulation are subject to cellular metaplasia, leading to various subtypes of OC. OC once initiated would then continue disseminating via the FT to the other gynaecological organs and peritoneal cavities. For Hysterectomy have added text -> According to Rice et al, the protective effect can be due to prevention of retrograde flow as mentioned prior. Another theory can be the “screening effect” i.e. surgeons removing possible pre-malignant lesions on direct visualization. Cutting off blood supply to the ovary would decrease the oestrogen production and hence reduced overall hormone exposure, implying another protective mechanism [68].
-
-
-
Table 1 additions to ASCO and USPSTF rows have been made.
-
Explicit AJ acronym in page 13 title has been made.
-
Additional comments have been made in Conclusion section -> This is also reflected economically, with mixed results from several studies analysing the cost effectiveness of screening programmes + Furthermore, the literature suggests a clear economic benefit in prophylactic measures within high-risk groups + However, this comes with the risks and long-term complications of the procedure, which is an important consideration given that the literature suggests increased clinical benefit and cost-effectiveness when carried out earlier in life. Whilst chemoprevention may be an alternative to young women who wish to preserve their fertility, there is currently limited evidence on its economic impact.
-
Reviewer 2 Report
In the manuscript from Ghose et al. authors clearly explain the biological background of hereditary ovarian cancer, what are today strategies for its prevention, and how much these strategies cost.
The review is well written and will be a useful resource in the field.
Comment: Mutations written in Figure 1. are hard to read.
Words do not have the same font size (e.g. Section 4.1. breast cancer).
Author Response
Dear Editor and Reviewers,
I am pleased to resubmit for publication the revised version of the manuscript ijerph-1926529, entitled “Hereditary Ovarian Cancer: Towards a Cost-Effective Prevention Strategy”.
Thankfully the reviewers provided us with a great deal of guidance, regarding how to better position the article. We are hopeful you agree that this revision will update our comprehensive review. All the comments have been addressed, as shown in the revised version of the manuscript, along with this point-by-point response to the reviewers' comments.
All corresponding are blue changes in the manuscript.
Reviewer #2:
In the manuscript from Ghose et al. authors clearly explain the biological background of hereditary ovarian cancer, what are today strategies for its prevention, and how much these strategies cost.
The review is well written and will be a useful resource in the field.
Comment: Mutations written in Figure 1. are hard to read.
Words do not have the same font size (e.g. Section 4.1. breast cancer).
Response:
Thank you for your appreciation. Your suggestions are much appreciated.
-
Figure 1 changes have been made.
-
From Introduction to References font is now uniform. Font Size = 10. Font Style = Palatino Linotype.
Reviewer 3 Report
In the review article entitled “Hereditary ovarian cancer: towards a cost-effective prevention strategy”, the authors summarized prevention strategies for hereditary ovarian cancer, international guidelines and cost-effective prevention strategies. It is very meaningful. Listed below are some of the minor concerns with this manuscript.
1. In the figure, the authors should add figure legend to briefly describe the figure.
2. In abstract, OC was defined as ovarian cancer. In the second sentence, the authors should use OC, but not ovarian cancer.
Author Response
Dear Editor and Reviewers,
I am pleased to resubmit for publication the revised version of the manuscript ijerph-1926529, entitled “Hereditary Ovarian Cancer: Towards a Cost-Effective Prevention Strategy”.
Thankfully the reviewers provided us with a great deal of guidance, regarding how to better position the article. We are hopeful you agree that this revision will update our comprehensive review. All the comments have been addressed, as shown in the revised version of the manuscript, along with this point-by-point response to the reviewers' comments.
All corresponding are blue changes in the manuscript.
Reviewer #3:
In the review article entitled “Hereditary ovarian cancer: towards a cost-effective prevention strategy”, the authors summarized prevention strategies for hereditary ovarian cancer, international guidelines and cost-effective prevention strategies. It is very meaningful. Listed below are some of the minor concerns with this manuscript.
1. In the figure, the authors should add figure legend to briefly describe the figure.
2. In abstract, OC was defined as ovarian cancer. In the second sentence, the authors should use OC, but not ovarian cancer.
Response:
Thank you for your appreciation. Your suggestions are much appreciated.
-
Figure legend is now more exhaustive -> Figure 1. The illustration shows distribution of gene mutations in ovarian cancer. Fig. 1a shows proportion of Sporadic (75%) and Hereditary (25%) Ovarian Cancer cases.
Fig. 1b expands details of HOC. The inner circle shows the proportions contributed by HBOC and Lynch Syndrome. The outer circle demonstrates the divisions shared by prominent genetic mutations (BRCA1, BRCA2, TP53) and mutation groups (Mismatch Repair Genes, Double Strand Break Repair Genes) corresponding approximately to the syndromic association in the inner circle. -
In the second sentence of the abstract Ovarian Cancer has now been changed to OC.